# Experimental Evaluation of Smartphone Accelerometer and Low-Cost Dual Frequency GNSS Sensors for Deformation Monitoring

**DOI:** 10.3390/s21237946

**Published:** 2021-11-28

**Authors:** Alexandru M. Lăpădat, Christian C. J. M. Tiberius, Peter J. G. Teunissen

**Affiliations:** Faculty of Civil Engineering, Delft University of Technology, 2628 CN Delft, The Netherlands; c.c.j.m.tiberius@tudelft.nl (C.C.J.M.T.); P.J.G.Teunissen@tudelft.nl (P.J.G.T.)

**Keywords:** deformation monitoring, GNSS, smartphone accelerometer, low-cost, measurement precision, noise characterization, GPS-based multipath correction

## Abstract

Smartphone accelerometers and low-cost Global Navigation Satellite System (GNSS) equipment have faced rapid and important advancement, opening a new door to deformation monitoring applications such as landslide, plate tectonics and structural health monitoring (SHM). The precision potential and operational feasibility of the equipment play an important role in the decision making of campaigning for affordable solutions. This paper focuses on the evaluation of the empirical precision, including (auto)time correlation, of a common smartphone accelerometer (Bosch BMI160) and a low-cost dual frequency GNSS reference-rover pair (u-blox ZED-F9P) set to operate at high rates (50 and 5 Hz, respectively). Additionally, a *high-rate* (5 Hz) GPS-only baseline-based multipath (MP) correction is proposed for effectively removing a large part of this error and allowing to correctly determine the instrumental noise of the GNSS sensor. Furthermore, the benefit of smartphone-based validation for the tracking of dynamic displacements is addressed. The estimated East-North-Up (ENU) precision values ($\hat{\sigma }$) of ±7.7, 8.1 and 9.6 $\frac{\mathrm{mm}}{s^{2}}$ are comparable with the declared precision potential (σ) of the smartphone accelerometer of $\pm 8.8\frac{\mathrm{mm}}{s^{2}}$. Furthermore, the acceleration noise shows only mild traces of (auto)correlation. The MP-corrected 3D (ENU) empirical precision values of ±2.6, 3.6 and 6.7 mm were found to be better by 30–40% than the straight-out-of box precision of the GNSS sensor, attesting the usefulness of the MP correction. The GNSS sensors output position information with time correlation of typically tens of seconds. The results indicate exceptional precision potential of these low-power-consuming, small-scale, affordable sensors set to operate at a *high-rate* over small regions. The smartphone-based dynamic displacement validation shows that GNSS data of a low-cost sensor at a 5 Hz sampling rate can be successfully used for tracking dynamic processes.

## 1. Introduction

Over the last decade, promising low-cost sensors have faced a rapid boom, making the analysis and prediction of damaging geophysical phenomena and abnormal structural responses possible at moderate cost for the applications of (for instance) landslide tracking and monitoring tall and slender civil engineering structures.

Depending on the application, sensors can be centered around two displacement types—dynamic or quasi-static displacements (see Figure 1). The major difference between a dynamic and quasi-static displacement lies in the reaction force time of the struck body and whether it is considerably large to produce an inertial force. Both displacement types can be present in geophysical and structural engineering applications.

Over the last two decades, high-rate micro-electro-mechanical system (MEMS) accelerometers and networks of high-end Global Navigation Satellite System (GNSS) receivers have represented the most useful and robust solutions for deformation monitoring.

Generally, MEMS accelerometers are used for capturing fast, dynamic responses at high rates (>1 Hz). It is known ([1,2]) that they work very well over high frequency ranges but perform poorly at sensing low-frequency, quasi-static movements. MEMS accelerometers are built with “sensitive axes, both in the plane of the device and perpendicular to that plane, enabling a three-axis accelerometer ensemble to be etched onto a single silicon chip of small dimension, exhibiting much greater shock tolerance than conventional mechanical designs” [3]. These sensors measure changes in capacitance corresponding to the acceleration in one, two or three directions of a proof mass and its fingers relative to a fixed ensemble of fingers attached to the frame of the accelerometer. Thus, the time series of accelerations can be measured, stored and analyzed to identify fine ground seismicity or structural vibrations, describing what GNSS sensors cannot resolve. Despite their size and affordable cost, MEMS accelerometers have relatively poor performance of navigation, influenced by higher frequency noise and weak response to static displacements. The studied inertial measurement unit (Bosch BMI160 [4]) is comprised of a 16-bit accelerometer and a low-power gyroscope.

On the other hand, GNSS sensors work based on the principle of wave travel time determination, where a minimum of four satellite-born radio carrier waves need to be acquired by a receiver on the ground to estimate its 3D position. Excluding modern high-rate (>1 Hz) GNSS equipment for seismology ([5,6]), such instruments are preferred for measuring quasi-static displacements ([1,7,8,9]). To facilitate this, GNSS systems can produce sub-centimeter precise relative position solutions based on the principle of Real Time Kinematic (RTK) differential positioning [10]. On account of the received GNSS carrier waves, carrier phase (CP), pseudorange and navigation data are used to estimate the 3D baseline, which is the relative position of the rover receiver with respect to the reference receiver, delivering such a solution in a kinematic way for every measurement epoch. Carrier wave cycle ambiguity resolution [11] is thereby crucial to attain sub-centimeter precise 3D baseline solutions. Every external influence on the reference or rover position will be reflected in the baseline solution as displacement from the average static state of the initial baseline, giving the chance to study permanent displacements over time exceeding the instrumental noise level. The studied GNSS receiver pair (u-blox ZED-F9P) is a low-cost multi band sensor “capable of centimeter precision” [12].

With the rapid advancement of such low-cost sensors recent research studies ([13,14]) highlight the increasing interest in affordable solutions for static deformation monitoring. There are many more interesting applications of these low-cost sensors, for instance, in measuring and monitoring vehicle dynamics ([15,16]). However, this study strictly focuses on the precision evaluation of static/permanent sensor set-ups targeting millimeter-precise geodetic applications.Notable achievements in the fields of plate tectonic, landslide or structural health monitoring have been documented, where some of the most recent and novel studies are mentioned below.

Allen et al. [17] demonstrated the effectiveness of smartphone accelerometers for earthquake early warning (EEW). Based on MyShake, their crowdsourcing seismic system, acceleration information from personal smartphones is clustered to report the expected approximate time and intensity of earthquakes in near real-time. Their results show that densely coupled inexpensive smartphone accelerometers can effectively detect the shakings of magnitude five and larger, produced at least 10 km away in the frequency range of 1 to 10 Hz. The authors demonstrate that such a crowdsourcing solution can compete with the performance of a high-end real-time EEW system by identifying a powerful earthquake within five seconds after its origin time.

To effectively explain deformation processes, instrumental precision and accuracy are two concepts of primordial interest. Odolinski et al. ([18,19]) studied the precision and ambiguity resolution performance of a low-cost single frequency differential GNSS system using multi-GNSS code and carrier phase data over short baselines. Their results reveal a sub-centimeter precision level, especially if the low-cost receiver is accompanied by a high-end antenna. On behalf of a tight positional dilution of precision (PDOP) constraint, the resulting ambiguity success rate of the low-cost GNSS system is comparable (almost 100%) to that of high-end receivers. Similar performance is achievable over a longer baseline (up to 9 km) if slant ionospheric delays are effectively modeled.

Several experimental studies have been carried out for the investigation of the precision and monitoring capability of low-cost single frequency GNSS systems of low-frequency landslide processes at a 1 Hz rate ([20,21]). The results show again that the noise level of such affordable systems situate within sub-centimeter level in the horizontal component and exceed 1 cm in the vertical direction. Furthermore, on behalf of classic differential positioning and statistical testing and the network Real Time Kinematic (NRTK) positioning method [22], manually induced permanent displacements were accurately recovered. Since the damaging impact of landslides exceed by far the centimeter level, both studies ([20,21]) highlight satisfactory results for landslide monitoring.

Manzini et al. [23] performed an extensive performance analysis on different GNSS antennae coupled with various low-cost single frequency receivers, to examine their potential for the SHM of civil engineering structures at a 1 Hz rate. Based on Post-Processing Kinematic (PPK) differential positioning, they show that a USD 350 low-cost single frequency GNSS system is capable of retrieving sub-centimeter horizontal displacements, centimetric vertical displacements and oscillations with frequencies up to 0.25 Hz exceeding 1 cm in amplitude, being more than enough for the monitoring of slender structures.

More recently, Hamza et al. [24] evaluated the precision performance of a low-cost dual frequency GNSS system by studying the variability in the 1 Hz rate CP residuals time series of two short baseline setups equipped with a low-cost and high-end GNSS sensor at the reference station location. The resulting residual statistics assess the horizontal noise level ($\hat{\sigma }$) of the differential GNSS system with about ±2 mm in case of both baseline setups and the vertical noise level with ±3 mm over the low-cost baseline and ±4.2 mm over the high-end baseline. It is noteworthy that the low-cost baseline setup performed better in the vertical plane due to the elimination of the antenna phase center offset (PCO) and variations (PCV) produced by the uncalibrated but identical low-cost GNSS antennae. The authors attest that by controlled horizontal displacement induction and statistical testing, both setups can effectively detect horizontal displacements larger than 1 cm.

Since all the aforementioned studies examine the precision and deformation monitoring potential of low-cost GNSS instruments at a 1 Hz rate at most, their performance at higher rates (>1 Hz) is of much interest. Therefore, the questions to be answered in this contribution read as follows: *“Are high-rate smartphone accelerometers and low-cost dual frequency GNSS receivers sufficiently precise to support deformation monitoring? Can a dynamic deformation monitoring process run with low-cost GNSS sensors benefit from smartphone-based validation?”*

Hence, this study addresses the empirical precision evaluation of a high-rate smartphone accelerometer and low-cost dual frequency GNSS receiver pair to support deformation monitoring applications. Furthermore, this study assesses a possible correlation in the high-rate observation time series, as a large positive correlation means that precision-wise, the information content of all of the observation time series as a whole is less than without any correlation.

For this purpose, we use a methodology to assess the mathematical observation model, for the empirical precision quantification, instrumental noise characterization and time (auto)correlation assessment of each sensor. Secondly, a double integration method is applied on data of an artificial SHM experiment run on a cantilever beam to validate the capability of the high-rate GNSS sensor of measuring moderate dynamic (beam) displacements in frequency.

The rest of this paper is structured as follows: Section 2 describes the experimental setup, needed materials and methodology for the evaluation of the mathematical observation models of both sensors. Section 3 presents the smartphone accelerometer and GNSS results in terms of empirical precision and time correlation performance. Furthermore, the construction, application and effectiveness of a GPS-only baseline based multipath (MP) correction is discussed. Finally, the GNSS based cantilever beam vibrations are validated against those observed by the accelerometer. Section 4 gives recommendations for the deployment of these sensors for precise deformation analyses. Final conclusions and answers to the research questions are given.

## 2. Materials and Methods

In this section, we describe our experiments, together with the mathematical model assessment methodology for the quality assessment of the two low-cost sensors.

### 2.1. Experimental Setup, Data Acquisition and Processing

To be able to measure and validate the response of a cantilever beam (CB) to wind load, a specific experiment named as the ‘Cantilever Beam’ experiment was set up. For this, an integrated SHM sensor network was developed consisting of a low-cost dual frequency differential GNSS system comprising identical dual frequency GNSS equipment at both baseline ends, a smartphone accelerometer and a weather station with an anemometer. The ensemble of sensors was installed in a rather clean environment regarding near-field effects and MP, in an open field near the town of Breda (φ = 51.6° north, λ = 4.7° east), the Netherlands, as seen in panel (a) of Figure 2. The primary sensors of the system were the smartphone accelerometer and the differential GNSS system. These recorded observations at a high rate (50 Hz for the smartphone accelerometer and 5 Hz for the GNSS) over four days from the 16 March until 20 March 2020. Each day is referred by their day-of-year (DOY): 076, 077, 078, 079 and 080 (The used (5 Hz) GNSS baseline estimate (.pos) files of day 077, 078 and 079, the MATLAB (.mat) data files consisting of the (50 Hz) steady MI and (50 Hz) AID experiment acceleration records were showed in Appendix A). Based on the recorded meteorological information, day 079 is attested as a day with very little wind when the rover antenna position was hardly influenced by any CB vibrations, making it perfect for the precision and time correlation analysis.

A second SHM experiment, named as the ‘Artificially Induced Displacement’ (AID) experiment, was performed on 11 June 2021 at the Delft University of Technology campus (φ = 52.0° north, λ = 4.4° east). A similar instrumental setup was used as with the CB experiment, excluding the weather station. The sensors recorded observations at a high-rate for about an hour and 20 min starting from 16:14:09 UTC. During this period, vibrations of the beam were initiated multiple times by pulling the beam to one side along the X (weak) axis using a rope and releasing it free to vibrate until rest.

On 5 July 2021, we collected a seven-hour-long set of acceleration observations starting from 00:19:37 UTC. This data set is denoted as the *steady MI* data set (MI is short for XiaoMI, the brand of the used smartphone with the Bosch BMI160 accelerometer). In this case, the smartphone was installed indoor on a horizontal leveled surface (table) at rest and continuously recorded acceleration observations at 50 Hz, being permanently connected to the local cellular network. The data set was influenced by a constant drift of the smartphone clock, which we correct for in Section 3.1.2. A preliminary calibration for assessing the non-orthogonality of the measuring axes was not conducted since our purpose is to perform an empirical analysis of the smartphone precision at measurement level. Omitting such a calibration does not affect our findings since systematic effects can be taken care of by a least squares (LS) polynomial removal approach (see Figure A1), leaving us with the measurement noise for further assessment.

The recorded steady MI acceleration observations and GNSS baseline solution data set from the fourth day (079) of the CB experiment allow us to empirically demonstrate the precision performance and assess the time correlation in the observations. The AID data were collected with the purpose of validating the capability of the low-cost GNSS sensor for capturing fast vibrations of the CB due to man-induced displacements.

Table 1 summarizes the common characteristics and modes in which the accelerometer and GNSS sensors operated during the CB, AID and steady MI experiment.

Regarding the data processing procedures, the raw smartphone accelerometer data are used without applying any processing algorithm. On the other hand, the GNSS measurement data are processed in PPK mode to output baseline solutions every 0.2 s (5 Hz). The processing configuration of RTKLIB’s software package [25] application program (AP), RTKNAVI, can be found in Table 2. For more information about the experiment, sensor installation, sensor use and data processing, please refer to [26].

### 2.2. Methodology

In the context of an estimation problem, the mathematical model represents a real world idealization describing the observables of the studied physical phenomenon, in our case, the steady state of a cantilever beam. Any displacement/deviation from this is interpreted as deformation.

A mathematical model consists of two parts (see Figure 3): a functional (FM) and stochastic model (SM). The FM relates the expectation of the observables ($\underline{y}$) to the unknown parameters of interest (*x*) subject of estimation, while the SM describes the quality of the observables through the observables’ variance–covariance matrix ($Q_{yy}$), their noise characteristics through the auto-correlation sequence (ρ(τ)) and possibly the observables distribution.

The two key steps for assessing a mathematical model are as follows:*FM assessment*—checking for the validity of the FM and unbiasedness of the observables;*SM assessment*—quality/precision assessment of the observables (in two steps):
-evaluation of the (empirical) precision ($\hat{\sigma }$) of the observables: straight-out of box (${\hat{\sigma }}_{a},\;{\hat{\sigma }}_{\hat{x}}$) and their noise level (${\hat{\sigma }}_{\hat{e}}$);-evaluation of observables (auto)correlation period ($t_{\rho (\tau )}$).

An illustration of the followed procedure for the assessment of the FM and SM of each sensor is presented in the second half of Figure 3 (see green and yellow panel).

The FM assessment is checking if the recorded acceleration observations (of the steady MI data set) and the sidereal East-North-Up (ENU) baseline estimates (of day 079) are fluctuating around their zero-mean and (true) mean value in the absence of any systematic bias or outlier.

On behalf of a valid FM, the SM can be then assessed to empirically describe the quality of the accelerometer and GNSS sensor. For this, the straight-out-of-box empirical precision of the acceleration and baseline observations is computed first. This precision estimate can incorrectly characterize the instrumental noise level by not accounting for erroneous effects acting as random measurement noise (such as far field background noise for the accelerometer and MP for the GNSS sensor). Hence, such erroneous effects are indirectly modeled and removed via an LS polynomial as to leave just random noise and correctly characterize the empirical precision of the two sensors. Next, a histogram gives a first impression on the statistical distribution of the observations. Finally, the presence of time correlation in the observation records is assessed by means of the empirical (auto)correlation function.

## 3. Results and Discussion

In this section, we present our FM and SM assessment results for the smartphone accelerometer and low-cost GNSS sensor. The results are discussed sensor-wise to draw conclusions on each sensor’s quality and suitability for deformation monitoring. Finally, the effectiveness of a high-rate baseline based MP correction is demonstrated, and a smartphone based validation approach for efficacious dynamic deformation monitoring is discussed and demonstrated as well.

### 3.1. Smartphone Accelerometer Results

#### 3.1.1. *FM Assessment Results*

With the smartphone accelerometer, the FM is constructed based on the assumption that a horizontally leveled accelerometer at rest should record (unbiased) zero average horizontal accelerations. In such a case, the recorded vertical acceleration samples are expected to be systematically influenced by the Earth’s gravitational acceleration, which needs to be subtracted from the records to output a zero-mean value. Taking into account that the mean values in the horizontal and vertical components in Figure A1 are marginally influenced by correctable far field background noise, minor linear offset errors and the Earth’s gravitational acceleration, our assumption on the validity of the FM can be accepted.

#### 3.1.2. *SM Assessment Results*

The straight-out-of-box precision of the accelerometer is characterized by the ENU empirical standard deviation values (${\hat{\sigma }}_{a}$) of a four-hour-long sequence of the steady MI acceleration data set of ±13.6, 8.1 and $9.9\;\frac{\mathrm{mm}}{s^{2}}$. Upon the subtraction of the black LS polynomial fit from the steady MI data set sequence (see Figure A1), the empirical precision of the smartphone accelerometer is characterized by the sample standard deviation values of the resulting ENU instrumental noise time series (${\hat{\sigma }}_{\hat{e}}$) in Figure 4 equal to ±8.0, 8.1 and $9.7\;\frac{\mathrm{mm}}{s^{2}}$. The LS polynomial worked as a correction for the linear offset and instrumental drift. Noteworthy is that after the LS polynomial subtraction, only the East empirical standard deviation value shows a considerable improvement with respect to the initial straight-out of box standard deviation value. Both the North and Up standard deviation values are similar to the straight-out of box standard deviation values. This is because the two data series were not influenced by any instrumental drift and the polynomial removing approach did not have anything to correct for, keeping the spread in the noise level of the accelerometer identical. Nevertheless, we decided to implement the polynomial removing procedure in each component to illustrate the general approach in accordance with Figure 3. Panel (a) of Figure 5 concludes a precision improvement of 41% in the East component and 2% in the Up component.

The aforementioned empirical precision values are comparable with the specified formal precision of 8.8 $\frac{\mathrm{mm}}{s^{2}}$ resulting from Equation (Equation 1). The specified formal precision value is a measure of the variability in the noise level (*e*) of the accelerometer operating at a sampling frequency (*f*) of 50 Hz based on a predefined spectral noise parameter ($n_{A,nd}$) of 180$\frac{\mu g}{\mathrm{Hz}}$. The spectral noise parameter value is defined based on in-lab calibration campaigns performed at a fixed temperature of 25 °C [4]: (1)$$\begin{array}{c} e=\sqrt{\frac{f}{2}}*n_{A,nd}*10^{-6}*9.81 \end{array}$$

Excluding the straight-out-of-box value of 13.6 $\frac{\mathrm{mm}}{s^{2}}$, deviations between the formal and empirical precision values are visible only in the Up component (see panel (a) of Figure 5), where the empirical values are approximately 10% poorer.

A preliminary inspection of the distribution of the noise sequences of the steady MI acceleration observations is considered. This is done by visually inspecting whether the (ENU) relative frequency histograms follow the bell shaped curve of the normal distribution’s probability density function (PDF). The visual check represents a first step in making a statement about the distribution of the observables. Results are shown in the top panel of Figure A3 in Appendix A. Little deviation between the histograms and the bell curves in the North (X) component is identified. There is no strong evidence against the normal distribution being a fair model for the steady MI acceleration observations, validating the foregoing principles behind the FM assessment and the empirical precision analysis.

Lastly, an (auto)correlation analysis is performed on a one-hour sequence of the steady MI data set to assess the time correlation in the observations. In the correlogram from the top panel of Figure 6, the (auto)correlation sequence (ρ(τ)) of the East noise sequence depicts an (auto)correlation period ($t_{\rho (\tau )}$) of 62 s. This reads as such since the behavior of the auto-correlation sequence converges to the 95% confidence interval of a purely white noise (AR0) process, settling after 3078 epochs and fluctuating very close to the upper limit of its confidence interval but without crossing it. For the North and Up components, the auto-correlation sequences cross and settle in the 95% confidence interval of an AR0 process after 550 and 137 epochs, respectively, corresponding to only 11 and 3 s of time correlation, respectively. The identified trace of (auto)correlation in the East component cannot be fully explained while the mild traces of correlation of the other two components are caused by the LS polynomial removing approach. These results are remarkable and affirm that a simple smartphone accelerometer is ready to output precise and nearly uncorrelated acceleration observations straight out of the box, giving access to any smartphone user for deliberate deformation monitoring.

### 3.2. Dual Frequency GNSS Results

#### 3.2.1. *FM Assessment Results*

For a static differential GNSS system with identical antennae at both baseline ends installed at similar heights, and particularly for a short baseline, the already well-known double-difference (DD) positioning model [10] can be assumed as a valid FM.

In this section, the sidereal ENU baseline time series of day 079 are used as observations. This is supported by the fact that day 079 is the only day of the CB experiment with very little wind and hence negligible impact on the baseline estimates. The results from Figure 7 indicate the zero-mean values of the instrumental noise attesting the unbiased, average static state of the rover and reference stations. However, a small deviation from zero is identified in the mean value of the estimated Up noise sequence. The estimated mean value of −0.1 mm situates below the noise level of any GNSS instrument, being negligible.

#### 3.2.2. *SM Assessment Results*

The straight-out of box precision of the GNSS system is characterized in Figure A2 of Appendix A by the (ENU) empirical standard deviation values (${\hat{\sigma }}_{\hat{x}}$) of the baseline time series equal to: ±3.7, 5.5 and 11.7 mm. Since the SM assessment is set to be deployed on the baseline time series of day 079, the observed harmonic variations in Figure A2 are likely caused by MP effects ([2,8]), which show up as time-correlated ’noise’ in the baseline estimates. The MP was (most likely) produced by the other instruments of the CB experiment surrounding the differential GNSS setup, the reflective property of the grassy field and the heights at which the GNSS antennae were installed. Hence, we likely incorrectly estimated the instrumental precision straight out of the box. In attempt to determine the instrumental noise level, we indirectly model the MP behavior by fitting hourly LS polynomials ([26,27]) through the baseline time series. The resulting hourly LS polynomial curves are glued together to define a high-rate, GPS-only, LS-based MP correction depicted by the black lines in Figure A2. The principle behind this correction relies on the ground track repeat cycle of the GPS constellation ([28,29]) of 23 h 56 min and 4 s. Hence, the black LS polynomial curves model the harmonic, time-varying but repeatable behavior of MP at the rover and reference station sites in 3D. Finally, these curves are subtracted component wise from the original baseline time series defining the (ENU) noise level time series from Figure 7, subject to the SM assessment. Upon subtraction, the empirical precision of the GNSS rover system in the East, North and Up components reaches ±2.6, 3.6 and 6.7 mm, respectively. Precision ’improvements’ relative to the straight-out-of-the-box precision of 30, 35 and 43% are observed (see panel (b) of Figure 5), indicating how one initially can incorrectly determine the instrumental precision from primary baseline estimates.

Subtracting the LS polynomial correction from the baseline time series of its origin is deemed insufficient for attesting the ground track repeatability property behind the MP correction. Furthermore, upon subtraction a certain degree of time correlation still stays in the noise sequence. This is elaborated in the following.

Following the outline of the precision performance analysis of the smartphone accelerometer in Figure 5, the empirical precision results are set side by side with the specified formal precision of 10 mm + 1 ppm. The formal precision was derived on behalf of multiple RTK measurement campaigns run over a one kilometer baseline with two identical patch antennae equipped with ground planes [12]. The observed variability in the noise level is here derived over a 15 m long baseline and indicates better precision relative to the formal value declared by the manufacturer. However, the empirical results should be treated as informative, highlighting the precision potential that can be reached over short baselines providing full sky visibility. The results show an exceptional precision potential of the GNSS module facilitating the monitoring of local deformation processes, such as landslide and SHM.

A preliminary inspection of the distribution of the noise sequences of the GNSS observations is performed (see bottom panel of Figure A3 in Appendix A). Based on the similarity between the shape of the derived relative frequency histograms and their corresponding normal PDF curves, one may tentatively assume that the normal distribution is a fair model for the CB experiment data set.

In the correlogram from the bottom panel of Figure 6, the red (auto)correlation sequence of a one-hour-long East noise sequence shows evident traces of (auto)correlation in the East baseline estimates, even after MP reduction. Such behavior is expected for sampling frequencies larger than the bandwidth of the phase-locked loop (PLL) mechanism (typically in the range of 5–15 Hz) of a GNSS receiver [10]. Knowing that the selected sampling frequency of 5 Hz is of similar order as the bandwidth of the receiver’s PLL, the identified time correlation cannot be explained by this certainty. The auto-correlation sequence firstly crosses the 95% confidence interval of an AR0 process after 349 samples, corresponding to an (auto)correlation period of 69.8 s. Surprisingly, its behavior does not settle within the black dashed 95% confidence interval in the bottom panel of Figure 6. Similar (auto)correlation period results of 117.4 and 45 s are obtained in the North and Up baseline components, respectively. The observed behavior and long (auto)correlation periods can be (likely) attributed to the LS polynomial removing approach and to the remaining MP post subtraction.

Figure 8 highlights the considerable traces of (auto)correlation in the (ENU) baseline estimates in contrast to the mild trace of (auto)correlation in the acceleration records. For each sensor, part of the time correlation is induced by the LS polynomial removing approach. However, in the case of the GNSS system, a large part of the identified (auto)correlation periods of 69.8, 117.4 and 45 s can be attributed to the incapability of the LS polynomial approach for substantially reducing MP. Therefore, a discussion on the effectiveness of a baseline-based GPS-only MP correction at reducing MP is presented in Section 3.3.

### 3.3. *Effectiveness of MP Corrections for Precise Deformation Monitoring*

In Section 3.2.2, an LS polynomial MP correction is applied on a high-rate baseline time series. This is insufficient to demonstrate the sidereal ground track repeatability property. In this section, we first compare the MP reduction effectiveness of the LS polynomial correction (${\hat{x}}_{p}$) with a high-rate baseline-based correction (${\hat{x}}_{m}$) using the baseline time series of day 079. Recall that both corrections rely on a static/permanent set-up of the GNSS receivers, hence they are useful for permanent (deformation) monitoring and not for kinematic applications such as automated driving. This is achieved by subtracting both corrections from a six-hour-long fragment of the baseline times series of day 078 ($\hat{x}$), the second day of the CB experiment, and comparing the results. The baseline time series fragment of day 078 spans from 00:00 to 06:00 UTC, when only a little wind was blowing along the narrow side of the CB producing minor impact to its static state. The two MP corrections and the data are aligned using the underlying sidereal time scale of the first day (077) of the CB experiment and assuming the repeatability of the GPS constellation. The resulting ENU noise time series from Figure A4 and Figure A5 of Appendix A show a random behavior freed from harmonic MP behavior, attesting to the proper functioning of each MP correction at first sight. When carefully comparing the two noise time series, the baseline-based noise band in Figure A5 shows even less sinusoidal behavior and looks even more random. Furthermore, the (ENU) empirical precision ($\hat{\sigma }$) of 1.9, 2.7 and 5.3 mm from the baseline based subtraction process is better (smaller) than the empirical precision of 2.3, 3.1 and 6.5 mm resulting from the LS polynomial subtraction process, concluding its higher effectiveness in reducing MP.

Secondly, to quantify the effectiveness of the baseline-based MP correction, the empirical standard deviations of the baseline based noise time series (${\hat{\sigma }}_{\hat{e}}$) are compared with the a-posterior formal values ($\sigma_{\hat{e}}$) resulting from the variance propagation law in Equation (Equation 2). With this variance propagation analysis, we take into account that the baseline-based MP correction (${\hat{x}}_{m}$) was defined from the data of another day (in this case, the next day, 079), and hence assume that the two time series are uncorrelated. Additionally, an MP reduction factor [27] is computed in Equation (Equation 3) by comparing the empirical standard deviations of the baseline-based noise time series with the ones of the MP-affected baseline time series (${\hat{\sigma }}_{\hat{x}}$). This process is run over the six-hour fragment of day 078. The results are summarized in Table 3.

When comparing the third and the fourth row component-wise, it is evident that the estimated empirical standard deviations of the baseline based corrected noise sequences are smaller than the expected formal standard deviation values from the variance propagation law. The variance propagation law ‘falsely treats’ the MP effect as random noise, which in fact is partially behaving also as a deterministic time varying bias [10], resulting in overestimated formal standard deviations. Thus, the reduction in the variability of the empirical results is an indication that the baseline-based MP correction was successful. The MP reduction values on the last row of Table 3 show that. for day 078, the MP correction was most efficient in the Up component by mitigating 50% of the inherent MP.
(2)$$\begin{array}{c} \sigma_{\hat{e}}=\sqrt{{\hat{\sigma }}_{\hat{x}}^{2}+{\hat{\sigma }}_{{\hat{x}}_{m}}^{2}} \end{array}$$
(3)$$\begin{array}{c} MP\;reduction=\left(1-\frac{{\hat{\sigma }}_{\hat{e}}}{{\hat{\sigma }}_{\hat{x}}}\right)*100\;\left[\%\right] \end{array}$$

The conclusion of the aforementioned comparison and the resulting MP reduction factors demonstrate the effectiveness of the GPS-only baseline based MP correction for performing more precise deformation campaigns over short baselines by correctly deriving the (empirical) precision potential of GNSS sensors. We demonstrate the use of such a simple MP correction developed to effectively work at a high sampling frequency of 5 Hz, making it accessible for more dynamic (GNSS-based) deformation monitoring. In the future, such corrections can be derived for multi-GNSS constellations over longer ground track repeat cycles.

### 3.4. Smartphone-Based Validation for Effective Dynamic Deformation Monitoring

After demonstrating the precision and feasibility of a smartphone accelerometer and low-cost dual frequency GNSS system, we validated the capability of the high-rate (5 Hz) GNSS system for measuring vibrations by using the accelerometer data.

The reverse transformation algorithm [8] was applied to derive displacements ($\tilde{x}$) from the acceleration measurements and validate the dynamic response captured by the GNSS sensor ($\hat{x}$). The workflow of this double integration algorithm is illustrated in panel (b) of Figure A6 and discussed in [8,26]. For our analysis, we choose a time period of 100 s (18:11:49–18:13:30 UTC) from the AID experiment. In Figure A7, the accelerometer- and GNSS-based dynamic displacement time series are plotted one on top of the other. The dynamic displacements match well in amplitude, being slightly larger for the high-rate (50 Hz) accelerometer. Looking at the horizontal axis, the internal quartz clock of the smartphone seems to run ahead of the more accurate GNSS receiver clock. This was expected knowing that the tolerance and stability of crystal oscillators is mainly dependent on manufacturing imperfections and environmental conditions (pressure, voltage or temperature changes) [30]. Hence, the smartphone clock drift was estimated to align the two displacement time series. For this, an empirical approach was taken, computing the drift between each pair of positive smartphone and GNSS-based vibration peaks and the elapsed time difference without synchronization of the smartphone based peaks. A (nearly) linear behavior of the smartphone clock drift was found with a rather poor clock stability of 0.01 $\frac{s}{s}$. This indicates that the smartphone clock drifts by 1 s every 100 s, making the two instruments not sample at the same time anymore already shortly after initiation. By accounting for the clock drift and offset, we align the responses of the two sensors and in particular consider one ’pull’ of the beam at 18:11:48 UTC. Panel (a) of Figure 9 shows a good match in amplitude and phase of the two filtered displacement time series, attesting to the capability of the low-cost GNSS receiver for dynamic deformation monitoring. However, we do not take into account on any methods for reducing the transient response of the low- and high-pass elliptical IIR filters. Hence, the filtered displacement time series ($\tilde{x}$ and $\hat{x}$) in Figure 9 are misleadingly warming up instead of showing a sudden jump to the maximum displacement produced by the ’pull’. Furthermore, the 1% discrepancy in the natural frequency from panel (b) results from the fact that the accelerometer time was not corrected for in the spectral analysis. Neglecting this, both sensors can output correct information about the structure’s natural frequency based on a Fast Fourier transform (FFT) analysis of the dynamic displacement data.

The reverse transformation algorithm serves as a useful validation method in dynamic deformation monitoring. This method can be very effective for post-processing deformation monitoring applications if the (smartphone) accelerometers are precisely time synced. Due to its dependency on filter designing and fine tuning, real-time monitoring is still a challenge. Its real-time applicability remains to be investigated in the future.

## 4. Recommendations and Conclusions

In the following, recommendations are outlined sensor-wise for the installation, efficient use and data analysis procedures in order to achieve a similar precision level to the presented results. Furthermore, important limitations from the CB experiment are highlighted to help users not repeat the same mistakes, giving a good and fair impression of the sensor’s operational feasibility. Based on the obtained results, a final conclusion on the sensor’s quality and suitability for dynamic deformation monitoring is expressed, answering the research questions.

### 4.1. Recommendations

Regarding the installation and efficient use of the smartphone accelerometer, we recommend to have the sensor always connected to the local cellular network or a GPS service in order to avoid time keeping errors that can cause fragmentation and missampling in the accelerometer data.

In the case of the GNSS sensor, in order to reach a high level of precision it is advisable to use identical antennae at both baseline ends to mitigate any antenna phase center (PC) errors. Otherwise, the rover antenna needs to be calibrated [31] by modeling its PCO and PCVs relative to the reference station. Furthermore, metallic ground planes for MP mitigation should be present along with the patch antennae in the differential GNSS setup. A high-precision performance may not be achievable for baseline lengths larger than 20 km (due to differential atmospheric delay errors ([18,19])) and for differential GNSS configurations surrounded by many high-reflective obstacles, in accordance with the technical specifications of the GNSS module [12]. Further research is required to assess the effectiveness of low-cost dual frequency GNSS sensors for deformation monitoring over large areas.

To practically limit time correlation in the baseline solution, we recommend setting up the low-cost GNSS sensor at sampling frequencies below the upper limit of the receiver PLL of typically 15 Hz [10].

Due to the limited length of the CB experiment (of four days) and the wind influence on the static state of the CB, we could define only a six hour MP correction based on the satellite ground track repeatability property using only GPS baseline observations. A multi-GNSS differential positioning approach ([18,19]) would further improve the precision, representing the next step to be considered. This goes hand in hand with the possibility of defining a multi-GNSS baseline-based MP correction over longer ground track repeat cycles.

Related to the post-processing MP techniques, we show that the baseline based [26] MP correction is more effective in mitigating MP than the LS polynomial-based correction. Regardless which of the two corrections, they come with the price of introducing some time correlation after application. In case of using an MP correction relaying on the ground track repeatability of the GPS constellation, a procedure for checking the repeatability of satellites and their geometry is desirable and should be addressed for deciding when to apply the correction. The first steps in this have been taken and documented in [26]. As a workaround, an elevation weighting approach has been documented in [18] to work at the code and CP levels.

Next to the steady MI experiment, the experiment was repeated several times with similar results for the smartphone accelerometer precision. Similar results were obtained after running an independent SM assessment using a different smartphone accelerometer. The precision results of the differential GNSS system fully rely on the long-enough sidereal CB experiment data. The effectiveness of the two instruments for dynamic deformation monitoring is supported by results from two SHM experiments.

### 4.2. Conclusions

It is incontestable that the sensor market has started to offer very cost-effective solutions for deformation monitoring. In the need of evaluating their performance, this paper empirically assessed the precision and operational feasibility of two promising and affordable sensor solutions for dynamic deformation monitoring set to operate at a high rate (50 and 5 Hz, respectively). The results show that both the smartphone accelerometer and the low-cost GNSS rover are able to output 3D acceleration and displacement information with $\frac{\mathrm{mm}}{s^{2}}$ and mm precision, respectively, at high rates, being more than sufficient for the monitoring of dynamic and quasi-static displacements.

The smartphone accelerometer produces valuable 3D vibration information with empirical (ENU) precisions ($\hat{\sigma }$) of ±7.7, 8.1 and $9.6\;\frac{\mathrm{mm}}{s^{2}}$ and almost inexistent time correlations of a few tens of seconds. The East component is an exception to this, giving slightly larger time correlations of up to 62 s for a (yet) unknown reason. However, the derived precision values are comparable with the declared formal precision (σ) of $\pm 8.8\;\frac{\mathrm{mm}}{s^{2}}$.

The dual frequency GNSS receiver outputs (ENU) displacement values with an empirical (ENU) precision ($\hat{\sigma }$) of ±3.7, 5.5 and 11.7 mm straight out of the box. The precision results are derived from a 15 meter baseline experiment. As expected, these values are smaller than the declared formal precision of ±0.1 m + 1 ppm since the later one was derived based on a tedious RTK calibration campaign run over a longer (one kilometer) baseline.

Furthermore, we show that if one tries to empirically asses the variation in the GNSS baseline position time series straight out of the box, one can incorrectly determine the precision of the GNSS sensor arriving at a too large standard deviation value. This reads as such since low-cost GNSS sensors, and effectively the baseline estimates, can be influenced by inherent multipaths. Therefore, we propose a high-rate (5 Hz), GPS-only, baseline-based MP correction to correctly determine the precision of the position time series of ±2.6, 3.6 and 6.7 mm. These new precision values are 30, 35 and 43% smaller than the straight-out-of-the-box precision values. However, the MP signal itself, when still present post-correction, may also cause a time correlation of typically tens of seconds. Lastly, a double integration method is applied on data from an artificial SHM experiment run on a cantilever beam to validate the capability of the high-rate GNSS sensor for measuring moderate dynamic (beam) displacements. Moreover, this method attests to the capability of both low-cost sensors to output precise frequency domain information by identifying the same natural frequency (1.36 Hz) of the cantilever beam. Hence, we conclude that the reverse transformation method serves as a useful smartphone-based validation method that can support GNSS-based dynamic deformation monitoring.

On behalf of the results, these sensors can be considered as affordable and precise options for regional landslide and SHM.

## Figures and Tables

**Figure 1 sensors-21-07946-f001:**
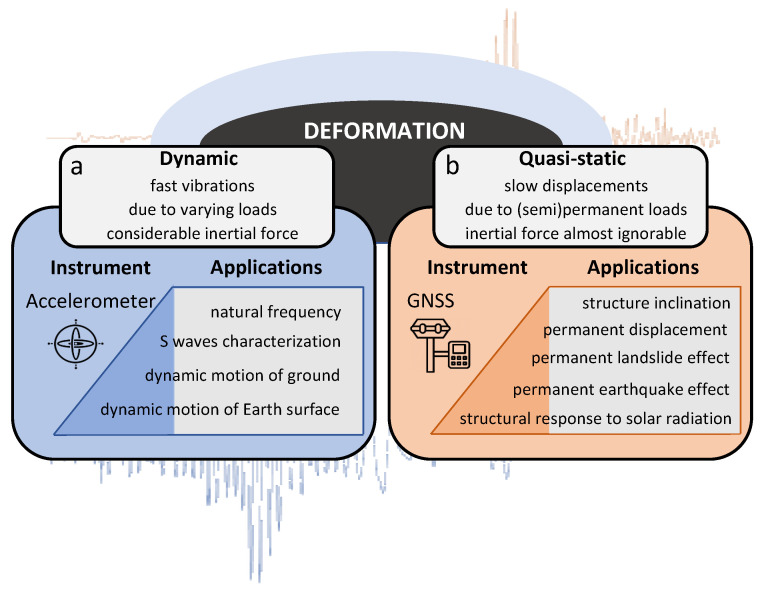
Displacement types: (**a**) dynamic and (**b**) quasi-static, their preferred tracking instruments and corresponding applications. Note the color choice for the two sensor types, blue for accelerometer and red for the Global Navigation Satellite System (GNSS).

**Figure 2 sensors-21-07946-f002:**
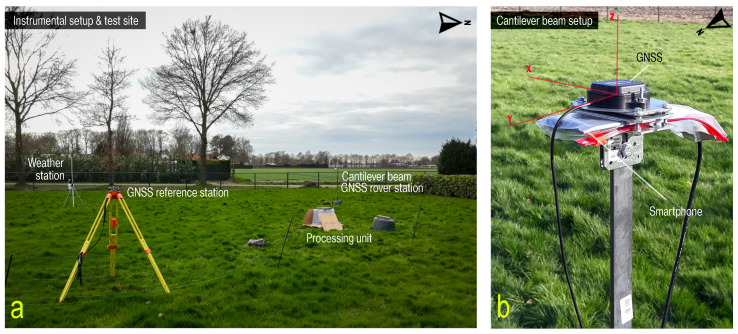
Instrument setup and test site of the ’Cantilever Beam’ experiment: (**a**) Overview of the instrument configuration viewed towards west. (**b**) Rover station sensors alignment at the beam’s top: low-cost GNSS antenna on top with a small metal disc ground plane and smartphone together with plastic cover protection below. The XYZ coordinate system in red coincides with the cardinal East-North-Up (ENU). We will restrict ourselves to the ENU notation in the following.

**Figure 3 sensors-21-07946-f003:**
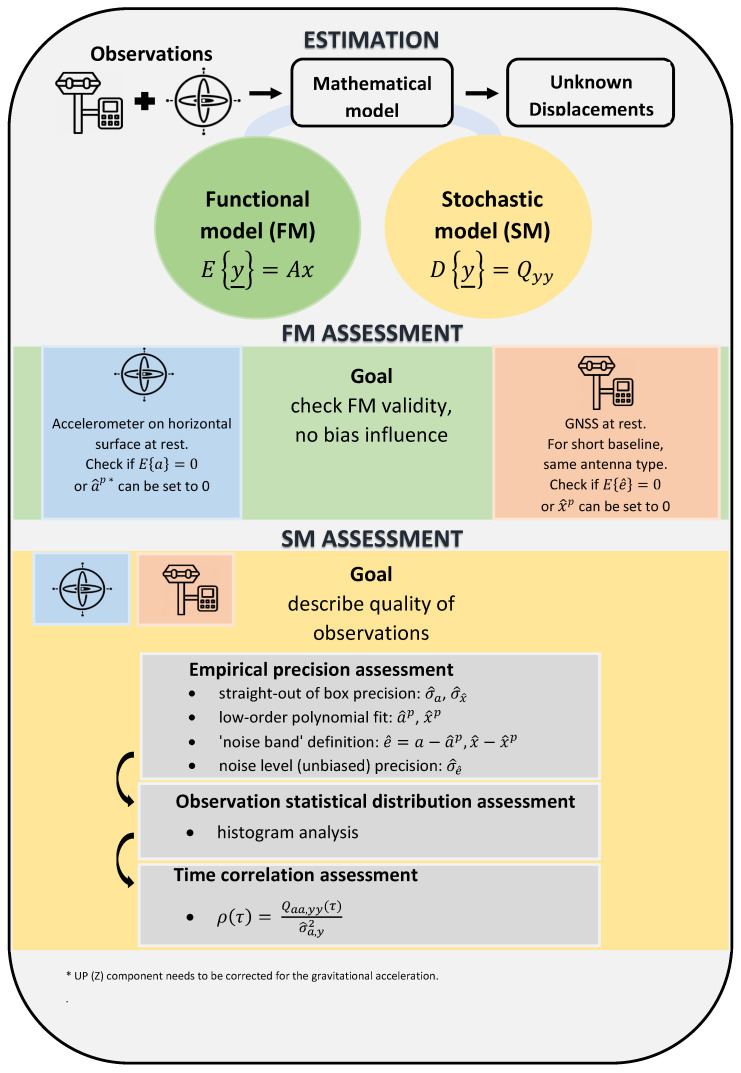
Methodology for mathematical model assessment: For both sensors, the procedure for (green) functional and (yellow) stochastic model assessment is illustrated. For explanation of the mathematical notations, please refer to the abbreviations panel at the end of this document.

**Figure 4 sensors-21-07946-f004:**
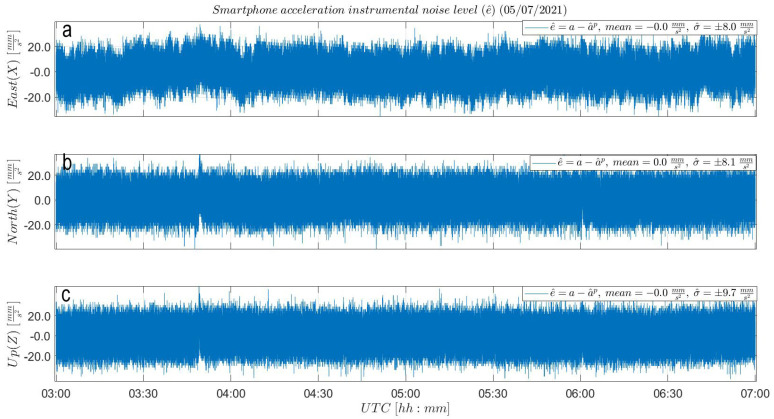
Smartphone accelerometer empirical noise ($\hat{e}$) analysis: (**a**) East, (**b**) North and (**c**) Up noise level of a four-hour-long sequence (after initiation) from the steady MI acceleration time series. Resulting ENU instrumental noise time series show a variability ($\hat{\sigma }$) of ±8.0, 8.1 and 9.7 $\frac{\mathrm{mm}}{s^{2}}$, respectively.

**Figure 5 sensors-21-07946-f005:**
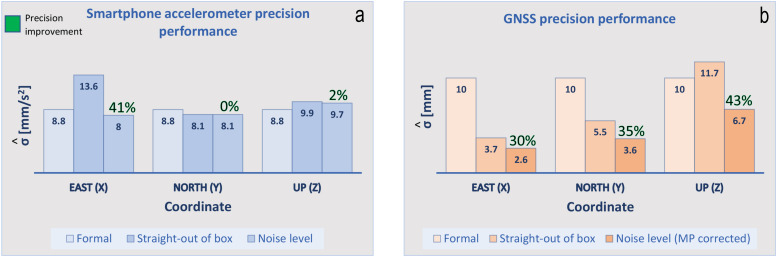
ENU precision results: (**a**) Smartphone accelerometer results indicate comparable precision relative to the specified formal precision value. The East straight-out-of-the-box precision is an exception due to accumulated drift error. Minor traces of precision degradation are visible in the Up component. Note that the formal precision value is derived based on a spectral noise parameter ($n_{A,nd}$) of 180 derived in specific laboratory conditions at a constant temperature of 25 °C [4]. (**b**) GNSS system-results show considerable discrepancies (precision ‘improvement’) between the straight-out-of-box and instrumental precision upon implementation of a GPS-only least squares (LS) -based MP correction.

**Figure 6 sensors-21-07946-f006:**
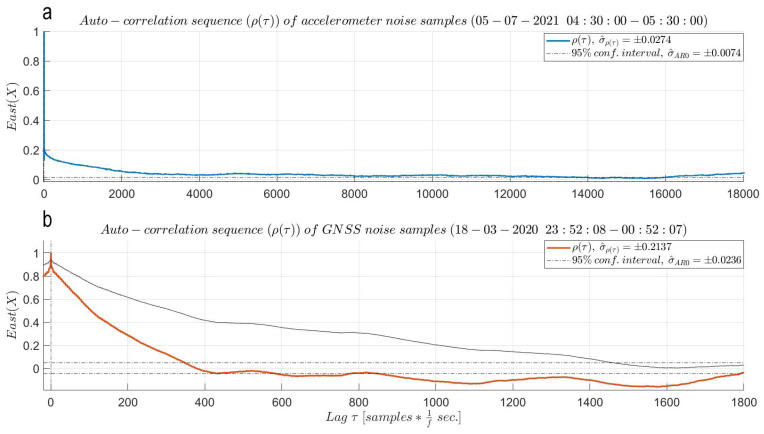
East (auto)correlation analysis: (**a**) Smartphone accelerometer—correlogram of random noise sequence converges close to the 95% confidence interval of an AR0 process, settling after 3078 samples and fluctuating close to its upper limit, corresponding to a correlation period of 62 s. The North and Up components show almost no time correlation, with correlation periods of 5 and 1 s, respectively. (**b**) GNSS system—red correlogram of random noise sequence enters into the 95% confidence interval of an AR0 process after approximately 349 samples followed by still some fluctuations. This corresponds to an (auto)correlation period of 69.8 s. The thin black correlogram is derived from the noise sequence (uncorrected for MP) resulting from the subtraction of the mean baseline value from the East baseline time series. The difference in time correlation between the two auto-correlation sequences attests to the reduction in MP on behalf of the LS polynomial subtraction. For the GNSS instrument, the North and Up components show similar behavior. Both (auto)correlation functions are plotted up to a lag at maximum equal to $\frac{1}{10}$ of the length of the selected one hour sequences.

**Figure 7 sensors-21-07946-f007:**
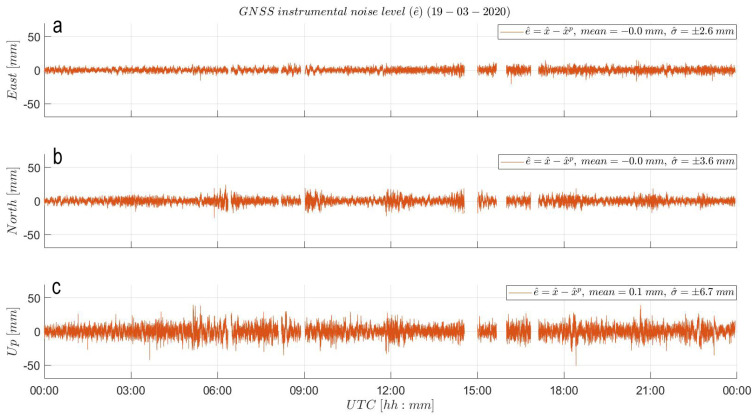
GNSS empirical noise ($\hat{e}$) analysis: (**a**) East, (**b**) North and (**c**) Up noise level of the baseline time series after LS polynomial subtraction. Resulting ENU instrumental noise time series show variabilities (${\hat{\sigma }}_{\hat{e}}$) of ±2.6, 3.6 and 6.7 mm, respectively. The gaps in data correspond to filtered out samples derived from float baseline solutions (rather than solutions with fixed ambiguity). Note the sidereal period of the time scale on the horizontal axis. The start and end point correspond to 18 March 2020 23:52:08 UTC and 19 March 2020 23:48:12 UTC, respectively, but for simplicity, they are plotted as 00:00:00 and 23:56:04 UTC on 19 March 2020.

**Figure 8 sensors-21-07946-f008:**
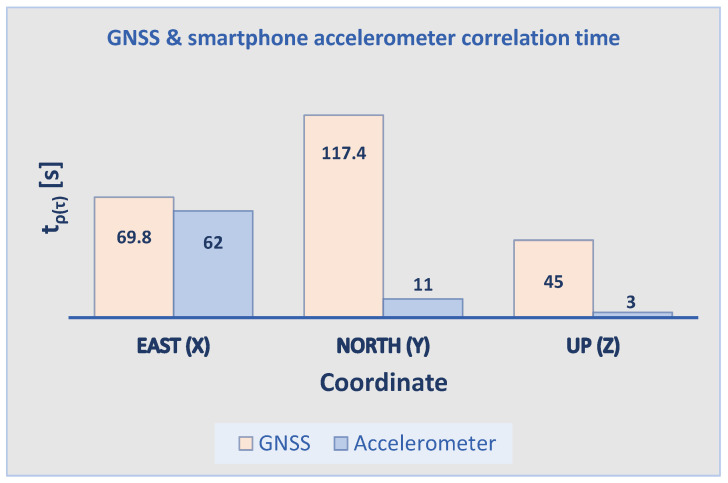
Comparison of correlation times in East, North and Up direction: The GNSS sensor shows an expected trace of correlation in all three components (due to LS polynomial removing and remaining MP), while the smartphone accelerometer shows mild traces of correlation; the East component is an exception to this, giving time correlation of up to 62 s for a (yet) unknown reason.

**Figure 9 sensors-21-07946-f009:**
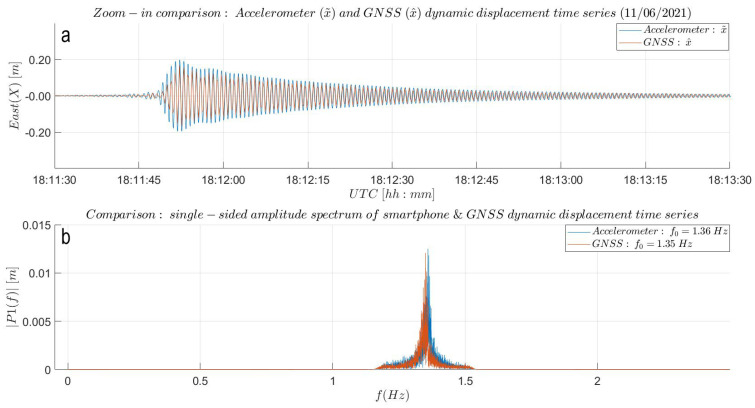
AID experiment results: (**a**) Zoom-in into Figure A7. Overlap of dynamic displacements. Smartphone accelerometer time series was shifted by 47.93 s to the right. Additionally, from the beginning of vibrations, the time series was shifted by 0.01 $\frac{s}{s}$ to the right to compensate for the smartphone oscillator in stability. The good match in amplitude and phase attests the capability of the low-cost GNSS instrument of dynamic deformation monitoring. Note the warming up effect of both displacement time series produced by the transient response of the IIR filtering process. Instead of this, a sudden ’pull’ to maximum was expected. (**b**) Natural frequency match without prior time correction. The identified natural frequency of the CB is of 1.36 Hz.

**Table 1 sensors-21-07946-t001:** Instrumental characteristics and modes of primary sensors in CB, Artificially Induced Displacement (AID) and steady MI experiment.

Sensors	GNSS	Accelerometer
Model	u-blox ZED-F9P + ANN-MB-00	Bosch BMI160 IMU, Xiaomi MI9 SE
Sensor	Multi-band GNSS receiver + antenna	Digital triaxial accelerometer and gyroscope
Size [mm]	17 × 22 × 2.4 + 60 × 82 × 22.5	2.5 × 3 × 0.8
Satellite constellations	GPS	-
Power consumption [W]	0.204 + 0.075 = 0.279	0.648
Operational mode	PPK	±8 g
Operational sampling rate [Hz]	5	50
Price [EUR]	229	4.49 chip, 420 smartphone integrated

**Table 2 sensors-21-07946-t002:** RTKNAVI processing configurations—PPK processing.

Setting	Option
Positioning Mode	Kinematic
Satellite constellations	GPS
Frequencies/Filter Type	L1+L2/Forward
Min Ratio to Fix Ambiguities	3
Elevation Mask [°]	15
Integer Ambiguity Resolution	Continuous
Output-Solution Format	E(X)/ N(Y)/ U(Z)-Baseline

**Table 3 sensors-21-07946-t003:** Baseline-based MP correction effectiveness on day 078.

day 078			
**Measure**	**East**	**North**	**Up**
${\hat{\sigma }}_{\hat{x}}$ [mm]	3.1	4.8	10.6
${\hat{\sigma }}_{{\hat{x}}_{m}}$ [mm]	3.1	4.9	10.7
${\hat{\sigma }}_{\hat{e}}$ [mm]	1.9	2.7	5.3
$\sigma_{\hat{e}}$ [mm]	4.4	6.9	15.1
MP reduction [%]	39	44	50

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
