# Peer review of "Experimental Evaluation of Smartphone Accelerometer and Low-Cost Dual Frequency GNSS Sensors for Deformation Monitoring"

_sensors, 2021, doi:10.3390/s21237946_

Round 1

Reviewer 1 Report

In this paper, through experiments,the accuracy of ENU is measured by Smartphone Accelerometer and low-cost Dual Frequency GNSS Sensors with multipath (MP) correction, respectively. And Smartphone Accelerometer and low-cost Dual Frequency GNSS Sensors with multipath correction can be used for deformation monitoring is concluded. In general, there are many problems, listed as follows:

  1. There are many types of formatting problems with incomplete charts in the article
  2. The format of the cited documents in the article is wrong, and the references are missing
  3. This article only verifies the actual accuracy of smartphone accelerometers and low-cost Dual Frequency GNSS Sensors with multipath (MP) correction for deformation monitoring, and lacks innovation.
  4. Verify that the actual accuracy of the smartphone accelerometer and low-cost Dual Frequency GNSS Sensors with multipath (MP) correction is not causally related to the conclusion that it can be used for deformation monitoring.
  5. The article lacks theoretical support, especially for the MP correction algorithm, lacks the introduction of the algorithm.

Therefore, we regret to reject the paper.

Reviewer 2 Report

Thanks to the authors for the interesting research results. However, while reading, several questions arose:  1. Lines 40-43: the authors indicate "These sensors measure change in capacitance corresponding to the acceleration in one, two or three directions of a proof mass and its fi ngers relative to a fi xed ensemble of fi ngers attached to the frame of the acceleromete." It is worth explaining the construction of the microaccelerometer used or giving a link to the source where it is shown. 2. Lines 47-48: the authors write "... where a minimum of four satellite born radio carrier waves need to be acquired by a receiver on ground to estimate its 3D position". And if this condition is not met (for example, in a mountainous area or a gorge), is it possible for this system to work? 3. Before starting the experiment, was the preliminary calibration of the used microaccelerometer performed on a special stand? This procedure is necessary for the correct interpretation of the data received from the sensor, as well as for assessing the non-orthogonality of its measuring axes (page 32, https://roboparts.ru/upload/iblock/1d0/1d0acb2968c1a5fca4d3b4ffb2c70bfb.pdf). 4. Lines 233-235: the authors make the assumption "With the smartphone accelerometer the FM is constructed based on the assumption that a horizontally leveled accelerometer at rest should record (unbiased) zero average horizontal acceleratio". On what basis is this assumption made? Has the parallelism of the used measuring axes of the microaccelerometer and the plane of the smartphone body been checked? 5. Figure 4 caption: "The aforementioned empirical precision values ​​are comparable with the speci fi ed formal precision of 8.8 mm / s2 resulting from Equation ??." What was the measurement range of the microaccelerometer during the experiment: ± 2g, ± 4g, ± 8g, ± 16g (its sensitivity depends on this)? (page 8, https://roboparts.ru/upload/iblock/1d0/1d0acb2968c1a5fca4d3b4ffb2c70bfb.pdf) 6. Caption to Figure 4: "The spectral noise parameter value is determined based on in-lab calibration campaigns performed at a fixed temperature of 25◦C". At what temperature outside was the experiment carried out? Could this affect the results obtained?

Reviewer 3 Report

The manuscript is of strategic importance because it traces the fiduciary limits of reliability and precision of the accelerometers present on smartphones for applications relating to civil engineering and geotechnics.

In the introduction, just for completeness of information in the manuscript, I would also add the widespread use that many researchers make to monitor driver behavior and vehicle dynamics.

I recommend the consultation this paper:

R. Vaiana, T. Iuele, V. Gallelli & D. Rogano (2017): Demanded versus assumed friction along horizontal curves: An on-the-road experimental investigation, Journal of Transportation Safety & Security, DOI: 10.1080/19439962.2016.1277290

Round 2

Reviewer 1 Report

In this article, the accuracy of East-North-Up (ENU) measured by the smart phone accelerometer and the multipath repaired low-cost Dual Frequency GNSS Sensors were obtained through experiments.It is concluded that the Smartphone Accelerometer and the multipath repaired low-cost Dual Frequency GNSS Sensors can be used for deformation monitoring. There are some problems, listed below:

  1. This article only verifies the actual accuracy of Smartphone Accelerometer and multipath repaired low-cost Dual Frequency were obtained for deformation monitoring, and lacks innovation.
  2.  The structure of the abstract of the article is messy, and the focus of expression is not prominent enough.
  3.  The results verified in this article show that the accuracy of the east component of the northeast celestial coordinate system is not high, and the reason is not given in the article, which cannot support the final conclusion.
  4.  The article lacks theoretical support, especially for the MP correction algorithm, lacks the introduction of the algorithm.
  5. Thus sorry to reject the manuscript
